# Plasminogen System in the Pathophysiology of Sepsis: Upcoming Biomarkers

**DOI:** 10.3390/ijms241512376

**Published:** 2023-08-03

**Authors:** Filomena Napolitano, Valentina Giudice, Carmine Selleri, Nunzia Montuori

**Affiliations:** 1Department of Translational Medical Sciences, University of Naples “Federico II”, 80138 Naples, Italy; filomena.napolitano@unina.it; 2Hematology and Transplant Center, University Hospital “San Giovanni di Dio e Ruggi d’Aragona”, 84131 Salerno, Italy; vgiudice@unisa.it (V.G.); cselleri@unisa.it (C.S.); 3Department of Medicine and Surgery, University of Salerno, 84081 Baronissi, Italy; 4Center for Basic and Clinical Immunology Research (CISI), WAO Center of Excellence, University of Naples “Federico II”, 80138 Naples, Italy

**Keywords:** biomarkers, sepsis, critically ill, suPAR, PAI-1, emergency department

## Abstract

Severe hemostatic disturbances and impaired fibrinolysis occur in sepsis. In the most serious cases, the dysregulation of fibrinolysis contributes to septic shock, disseminated intravascular coagulation (DIC), and death. Therefore, an analysis of circulating concentrations of pro- and anti-fibrinolytic mediators could be a winning strategy in both the diagnosis and the treatment of sepsis. However, the optimal cutoff value, the timing of the measurements, and their combination with coagulation indicators should be further investigated. The purpose of this review is to summarize all relevant publications regarding the role of the main components of the plasminogen activation system (PAS) in the pathophysiology of sepsis. In addition, the clinical value of PAS-associated biomarkers in the diagnosis and the outcomes of patients with septic syndrome will be explored. In particular, experimental and clinical trials performed in emergency departments highlight the validity of soluble urokinase plasminogen activator receptor (suPAR) as a predictive and prognostic biomarker in patients with sepsis. The measurements of PAI-I may also be useful, as its increase is an early manifestation of sepsis and may precede the development of thrombocytopenia. The upcoming years will undoubtedly see progress in the use of PAS-associated laboratory parameters.

## 1. Introduction

Sepsis has been defined by The Third International Consensus Definition for Sepsis and Septic Shock (Sepsis-3) as “organ dysfunction caused by a dysregulated host response to infection”, highlighting the crucial role of the innate and adaptive immune systems in the development and progression of the syndrome. In addition, septic shock has been defined as “subset of sepsis in which particularly profound circulatory, cellular, and metabolic abnormalities are associated with a greater risk of mortality than with sepsis alone” [1].

Despite great progresses on experimental and clinical research in recent decades, sepsis is a condition among hospitalized patients that carries a high risk of morbidity and mortality. In fact, mortality rates range from 15% in patients with sepsis without shock to 56% in patients with sepsis and with shock and depend on several factors, such as age, female gender, and comorbidities [2]. The degree of organ dysfunction is the strongest independent predictor of mortality, with mortality rates ranging from 10% with the failure of one organ to nearly 80% with the failure of four or more organs [1,3]. Importantly, survivors are at an increased risk of death or a reduced health-related quality of life after discharge from the hospital.

Historically, the definition and diagnosis of sepsis have undergone considerable changes. Before 2016, the systemic inflammatory response syndrome (SIRS) criteria were used in clinical practice. Thus, sepsis was defined as the fulfilment of two or more SIRS criteria (fever, tachycardia, hyperventilation, leucocytosis) in association with suspected infection. However, the SIRS criteria are sensitive to sepsis but not specific, as most infected patients will satisfy at least two SIRS criteria without ever developing organ dysfunction and/or circulatory abnormalities [3].

The Sepsis-3 criteria have introduced the system of the Sequential Organ-Failure Assessment (SOFA) score and the quick (q)SOFA to define sepsis. The score grades abnormality by organ system and accounts for clinical interventions. An increase in the SOFA score ≥ 2 indicates organ dysfunction. However, laboratory parameters are needed for full computation [1].

The early recognition of sepsis and organ dysfunction is crucial to improve survival, but the poor understanding of the pathophysiology reflects the gaps in the early diagnosis and the lacks in standardized biomarkers.

An ideal biological marker for sepsis can be used in early diagnosis, risk stratification, and treatment monitoring. In recent years, many biomarkers have been studied for their potential role in sepsis diagnosis, but the translation of research results into clinical practice presents many difficulties. Various inflammatory markers that are currently in use, including procalcitonin and C-Reactive Protein (CRP), have been evaluated in clinical and experimental studies. In particular, procalcitonin exhibits a low positive predictive value and is downregulated in viral infections [4,5]. Presepsin is specific for bacterial infections, especially Gram-negative infections, and gives false-positive results in some conditions, such as renal failure [6]. Nonetheless, presepsin represents a good accuracy for the diagnosis of sepsis compared to patients affected by systemic inflammatory diseases. Pentraxin-3 (PTX-3), which plays a crucial role in the early phase of inflammation by activating the complement cascade, was identified as a marker of sepsis severity and exitus [7]. Song et al. found that IL-6 has superior diagnostic and prognostic value compared to PTX-3, CRP, and procalcitonin [8]. CRP is considered a reliable biomarker compared to white blood cells (WBCs), but only when it is used in combination with severity clinical score [5].

Given that the scientific community has focused on inflammation as a unique pathway involved during the disease process, all investigated analytes are inflammatory mediators. Recently, it has been shown that anti-inflammatory mediators are also associated with multiple clinical manifestations of sepsis, such as multi-organ failure and vessel damage [9].

Understanding the pathogenesis and the complex interaction between different pathways occurring during sepsis is urgently needed to identify novel specific biomarkers to improve the prognosis and decision-making process.

## 2. Pathophysiology of Sepsis: An Intricate Interplay between Inflammation, Coagulation, and Endothelial Dysfunction

Infection and sepsis induce complex derangements in many systems, such as the immune system and the coagulation cascade. Knowledge about the pathophysiology of sepsis has greatly improved in recent years, and novel diagnostic tools have been proposed for the transfer to clinical practice. In this section, we would like to reconstruct the salient stages of the pathophysiological mechanisms that occur during sepsis and to advance the questions that are still open.

### 2.1. Hyper-Inflammation and Immune Suppression in Sepsis

Traditionally, sepsis was defined as a hyper-inflammatory syndrome, often characterized by a fast “cytokine storm”. In 1992, the term “systemic inflammatory response syndrome” (SIRS) was introduced, and sepsis was indicated as a “systemic inflammatory response syndrome to infection” [10]. If SIRS occurred without infection, the condition was defined only as SIRS; an infection without SIRS could not be considered as a sepsis. The current concept of sepsis, formulated by the Sepsis 3 task force, provides for the omission of SIRS [11]. The results of one study have shown that the presence of SIRS did not influence the overall prognosis, and the requirement of two or more SIRS criteria for the diagnosis of severe sepsis excluded a sizable group of patients in the ICU with infection and organ failure [12].

However, the first pathogenetic event that characterizes sepsis is excessive inflammation. After infection, the invading pathogen elicits the innate immune response via the recognition of pathogen-derived molecular patterns (PAMPs), and/or endogenous host-derived danger signals (DAMPs) that are the starting signals [13]. These molecules are recognized by specific pattern recognition receptors (PRRs), including Toll-like receptors, expressed on the cells of the innate immune system. The translocation of nuclear factor-kB (NF-kB) into the nucleus induces the activation of genes encoding cytokines that are crucial in triggering an excessive inflammatory reaction, as observed in sepsis. The typical pro-inflammatory cytokine panel of sepsis includes TNF-α, IL-1β, IL-12, and IL-18 [14].

A hallmark of sepsis-associated hyper-inflammation is the uncontrolled activation of complement, which can cause damage to tissues and organ failure. The final step of complement activation results in the release of anaphylotoxins, C3a and C5a, which exert their inflammatory effects by activating leukocytes, platelets, and endothelial cells. The blockade of C3a and C5a signaling represents an exciting therapeutic strategy that Mollnes and Lang define as a promising “upstream approach” before the inflammation becomes irreversible [15].

The activation of neutrophils causes the release of neutrophil extracellular traps (NETs) in order to limit the spread of infection and promote pathogen clearance. NETs are characterized by decondensed nuclear chromatin associated with proteins, including myeloperoxidase and elastase. Although the release of NETs is crucial for defense against pathogens, the uncontrolled release of NETs is associated with inflammation, tissue damage, and autoimmunity [16].

There is a debate on the timeline of immunological alterations. Recent studies have shown that both inflammatory and anti-inflammatory responses occur simultaneously in sepsis. The model proposed by Hotchkiss et al. indicated that during the early phase of sepsis, both pro-inflammatory and anti-inflammatory pathways are activated, but the pro-inflammatory response predominates [17]. It is conceivable that a balanced response could be crucial in infection control and e protection from organ dysfunction.

During the first 2–4 days, leukocytosis with significant increase in monocytes, neutrophils, and circulating pro- and anti-inflammatory cytokines has been observed. The condition of leukocytosis is followed by a state of lymphopenia characterized by a decreased number of CD4^+^, CD8^+^ T cells, and innate immune cells, especially natural killer (NK) and dendritic cells (DCs). The absolute number of leukocytes returns to normal after a month for most patients. However, surviving patients exhibit a long-lasting state of immunosuppression referred to as “immunoparalysis” and/or “immune exhaustion” [18,19]. This state, characterized by the apoptotic depletion of immune cells, is accompanied by increased viral replication/reactivation and susceptibility to secondary infections. In the later stages of sepsis, the dysregulation of the immune system progresses into an anti-inflammatory state. Cytokines such as TNF-α and IFN-γ are reduced, and cytokines such as IL-4, IL-10, and IL-37 are upregulated [20]. In particular, IL-10 may aggravate the state of immunosuppression by inhibiting the proliferation of CD4^+^ T cells, whereas IL-37 is closely related to the severity of sepsis-induced immunosuppression by suppressing the pro-inflammatory cytokine release from monocytes and neutrophils [21].

Sepsis-associated immunosuppression is supported by the data obtained from the studies of regulatory T (T_reg_) cells. The number of T_reg_ cells is increased in patients with sepsis, thus contributing to impaired T cell function. In an experimental model of sepsis, the blockage of T_reg_ cells improves immune response [22].

Finally, very little is known about the mechanisms underlying inflammation and immunosuppression associated with sepsis. Therefore, classical diagnostic and/or prognostic biomarkers commonly used in inflammatory diseases are not able to identify the condition of sepsis promptly and appropriately.

### 2.2. Coagulation Abnormalities in Sepsis

Most patients with sepsis in Intensive Care Units (ICUs) were diagnosed with sepsis-induced coagulopathy, also known as sepsis-induced disseminated intravascular coagulation (DIC). The mortality rate in sepsis significantly increases when DIC develops [23]. The main pathogenetic aspect of DIC is represented by the persistent activation of coagulation, but other aspects of the pathogenesis differ considerably based on the underlying disease, or rather sepsis, acute leukemia, and solid cancers. The typical trait of sepsis is suppressed-fibrinolytic-type DIC, with the severe activation of coagulation but a mild activation of fibrinolytic cascade [24].

Several studies have highlighted the extensive crosstalk between inflammation and coagulation as a pathogenetic pathway in sepsis-associated coagulopathy. The term “immunothrombosis” has been introduced to denote the marked correlation between the coagulation cascade and the inflammatory response [25]. Indeed, pro-inflammatory cytokines, including IL-6 and TNF-α, are able to activate the coagulation cascade and to downregulate anticoagulant system in sepsis [26].

The tissue factor is the initial trigger for coagulation activation and thrombin generation in sepsis. The tissue factor is located in the vascular endothelium; its exposure to circulation after endothelial damage leads to the activation of the extrinsic coagulation pathway. The tissue factor is also present in inflammatory cells, including monocytes, which are activated through PRRs by PAMPs and DAMPs, in the case of infection. Activated cells release extracellular vesicles that express procoagulant tissue factor and phosphatidylserine on their surface. Thus, the tissue factor is released into the circulation, and the extrinsic coagulation pathway is activated [27]. In experimental models of sepsis, the inhibition of the tissue factor prevents organ failure and mortality, probably not only attenuating the coagulation but also the inflammatory response [14].

The anticoagulant pathways, represented by tissue factor pathway inhibitor (TFPI), anti-thrombin, and the protein C system, are compromised during sepsis and promote thrombus formation [27].

Sepsis is characterized by the increased activation of platelets. The involvement of platelets in organ failure is due to platelet crosstalk with immune and endothelial cells during sepsis [28].

The link between NETs and thrombosis has been established. Several components of NETs, including tissue factor, the contact system, DNA, histones, and possibly elastase present on NETs, are all recognized activators of coagulation. Furthermore, NETs have been found to serve as a scaffold for activated platelets and red blood cells that bind to NETs through fibronectin, fibrinogen, and the von Willebrand factor [29].

### 2.3. Endothelial Dysfunction in Sepsis

Endothelial dysfunction is a central event in the pathogenesis of sepsis. It leads to multi-organ failure by enhancing vascular permeability, coagulation system activation, promoting tissue edema, and compromising the perfusion of vital organs.

In physiological conditions, the glycocalyx, a heparan-sulfate-rich layer of glycosaminoglycans and proteoglycans, maintains local selective permeability and a high reflection capability for albumin and is critically involved in leukocyte trafficking. During sepsis, qualitative and quantitative alterations in glycocalyx occur. In response to DAMPs and PAMPs, the vascular barrier is impaired due to glycocalyx breakdown, endothelial cell apoptosis, and junction protein dysregulation [30].

Under normal conditions, the endothelium contributes to homeostasis through the expression of antithrombotic and anti-inflammatory molecules. In sepsis, the endothelium is activated directly by PAMPs or indirectly by NETs and pro-inflammatory cytokines, including TNF-α, IL-6, and IL-1 [31].

Normally, the endothelium exerts anticoagulant properties. In sepsis, endothelial cells contribute to the hypercoagulable state through the decreased expression of thrombomodulin and heparan sulfate on the cell membrane and the increased expression of the tissue factor [13].

An important role in endothelial cell contraction and vascular leakage has been assigned to Formyl Peptide Receptors (FPRs). FPRs have been identified as a subfamily of G-protein coupled receptors. They are expressed at high levels on leukocytes and mediate cell chemotaxis. It has been demonstrated that FPRs stimulation leads to sepsis-like symptoms, including vascular leakage, thus contributing to the loss of control of vascular tone and edema in sepsis [32].

All the alterations in the endothelial homeostasis determine a shift to a pro-oxidant state, characterized by the abnormal production of reactive oxygen species (ROS) and reactive nitrogen species (RNS), and a loss of antioxidant systems. Joffre et al. have recently introduced a concept of a “vicious circle”, whereby the endothelium is both a source and a target for oxidative stress in sepsis. The disbalance of the redox signaling engages endothelial cells toward apoptosis, thus dramatically inducing endothelial dysfunction. Finally, endothelial cell death involves a loss of structure and membrane properties, which contributes to sepsis-induced permeability, capillary protein and fluid leakage, and therefore organ injury [33].

## 3. The Plasminogen Activation System as Key Player in the Pathogenesis of Sepsis

The plasminogen activation system (PAS), also known as the fibrinolytic system, is an enzymatic cascade involved in the degradation of preformed fibrin clots into soluble peptides. This system consists of serine proteases, activators, and inhibitors that finely control the formation of plasmin from its inactive precursor plasminogen [34].

Upon activation of the coagulation, fibrinogen is converted to fibrin, which is further stabilized via fibrin cross-linking by FXIIIa [35]. The activation of plasminogen into plasmin ensures the process of fibrin clot breakdown. However, plasmin is a broad-spectrum serine protease; in addition to well-characterized proteolytic activity on fibrin, it degrades various ECM components, such as laminin and fibronectin, and also activates procollagenases to active collagen-degrading enzymes [36].

The reaction of the conversion of plasminogen into plasmin is performed by specific activators. The most active plasminogen activator is tissue-type plasminogen activator (tPA), mainly secreted from endothelial cells in response to various stimuli, including thrombin activity, shear stress, histamine, and bradykinin. The second plasminogen activator is the urokinase plasminogen activator (uPA), which is known to have numerous functions beyond its involvement in fibrinolysis. It is synthesized by numerous cell types, including leukocytes, macrophages, tumor cells, and fibroblasts. Importantly, tPA requires binding to fibrin to mediate efficient plasminogen activation, whereas uPA is fibrin-independent and activates plasminogen in solution or when associated with its cellular receptor uPAR [37].

uPAR is a GPI-anchored cell membrane receptor composed of three homologous domains (DI, DII, and DIII), expressed on the surface of fibroblasts, endothelial cells, immune cells, and other tissue [38]. Beyond its function of locating the proteolytic activity of uPA to the cell surface, uPAR is considered a versatile receptor that elicits a plethora of cellular responses that include cellular adhesion, differentiation, proliferation, and migration. To perform these activities, uPAR cross-talks with different molecular partners on the cell surface, including vitronectin, integrins, growth factor receptors, and N-Formyl Peptide Receptors (FPRs) [39,40,41]. Interestingly, soluble forms of this receptor (suPAR) can be detected in body fluids (blood, serum, plasma, urine, saliva, cerebrospinal fluid, and pleural fluid) and has been suggested as a novel biomarker for triage and prognostication in the emergency department [42]. suPAR was first described in the 1990s as a marker of cancer progression as well as a marker of several infectious diseases. Since then, suPAR has been studied for its clinical, diagnostic, and prognostic potential [43].

The plasminogen activators, in turn, are regulated by specific inhibitors that include plasminogen activator inhibitors (PAI-1 and PAI-2). Plasmin is negatively regulated by α2-antiplasmin and α2-macroglobulin [44].

Another important regulator of fibrinolysis is thrombin-activatable fibrinolysis inhibitor (TAFI), also known as carboxypeptidase R, which removes C-terminal lysine binding sites for tPA and plasminogen from fibrin, decreasing the ability of fibrin to stimulate fibrinolysis [45].

In addition to its prominent role in fibrinolysis, PAS has been widely studied for its role in various physiological processes such as wound healing, cell signaling, and extracellular matrix (ECM) degradation. PAS dysregulation has been linked to the development and progression of diseases with an inflammatory component [46].

Inflammation, coagulation abnormalities, and endothelial dysfunction are the hallmarks of sepsis. The main evidence of the involvement of PAS in each of these pathways is reported below. However, the mechanisms that modulate fibrinolysis in sepsis remain incompletely understood.

### 3.1. Links between Fibrinolytic Mediators and Inflammation in Sepsis

Uncontrolled plasmin generation is associated with an abnormal inflammatory response. In a model of acute inflammation, Napolitano et al. have hypothesized that PAS is at the interface between complement and kallikrein systems. During acute inflammation, PAS and the kallikrein pathway generate an auto-amplification loop that provokes the abnormal production of the bioactive peptide bradykinin. Plasmin is a trigger for the release of pro-inflammatory bradykinin, but at the same time bradykinin itself stimulates endothelial cells to release tPA, and plasmin in excess is formed by this mechanism. Secondly, plasmin can modulate complement activation, thereby generating anaphylatoxins C3a and C5a. This mechanism induces mast cell degranulation, which results in histamine-mediated vasodilation and an increase in vascular permeability [34]. This intricate mechanism can also occur in the early stage of sepsis, thus preparing the ideal microenvironment for the development of severe disease. There are controversies among the scientific community regarding the role of PAS in the pathogenesis of sepsis. Despite the classical view of the fibrinolytic system as an inducer of pro-inflammatory pathways, accumulating evidence highlights the pro-resolving and anti-inflammatory capacity of fibrinolysis during inflammation.

Lower plasminogen levels are associated with the severity of sepsis. Upon a given infection, a coagulation cascade is rapidly activated in order to contain the spread of the microorganism. The activation of coagulation also triggers fibrinolysis to control excessive coagulation. The reduction in plasminogen levels observed in sepsis may be due to several mechanisms. Uncontrolled inflammation leads to the excessive consumption of pro-fibrinolytic components over time and the switch to increased production of fibrinolysis inhibitors. Importantly, the DNA-bound elastase of NETs is protected from inhibition by plasma antiproteases and sustains its ability to degrade plasminogen [47]. This mechanism can contribute to reducing plasmin formation, thereby compromising the fibrinolytic processes. In addition to these mechanisms, the impairment of liver biological function due to persistent systemic inflammation may contribute to decreased plasminogen levels [48].

Vago et al. have suggested that PAS is a crucial protective player during sepsis that reduces NETs and systemic inflammation [48]. Neutrophil apoptosis induced by plasmin(ogen) may be the underlying mechanism of decreased NET release [49]. Low levels of plasminogen were found in severe COVID-19 patients and therefore proposed as a negative prognostic factor in COVID-19 [50]. As opposed to the above data, a detrimental role of plasmin(ogen) during sepsis has been found. In particular, Guo et al. suggested that reducing or removing functional plasmin in mice leads to a higher survival rate during sepsis due to impaired cytokine production [51].

The role of tPA in the pathogenesis of sepsis is still controversial. Although increased levels of tPA are common in patients with sepsis and are associated with a worse outcome [52], Renckens et al. have demonstrated that endogenous tPA serves a protective role during *Escherichia coli* peritonitis that is independent of plasmin [53].

The uPA/uPAR system mediates inflammatory and immune responses and is involved in the pathogenesis of fibrotic diseases, such as systemic sclerosis and rheumatoid arthritis [39,41,54]. Even though the role of uPA in sepsis remains largely unknown, it is a useful mediator in inflammation for the activation of T cells and also acts as an antibiotic agent in a mouse model infected with *Staphylococcus Aureus* [55].

uPAR signaling is linked to inflammation and innate immune activation. Importantly, uPAR act as a part of PRR signaling to different PAMP and DAMP molecules. It has been demonstrated that the inflammatory reaction induced by TLR4 in response to LPS is reinforced by uPAR [56]. The complement C5a is the early event in the initiation of sepsis, as it induces a severe inflammatory response. The inactivation of uPAR blocks the sepsis-induced activation of C5a [57]. Furthermore, uPAR^−^/^−^ mice infected with *B. pseudomallei* exhibited a reduced ability to recruit the neutrophil towards the pulmonary environment, thus enhancing bacterial growth and dissemination [58].

PAI-1, the main inhibitor of fibrinolysis, exerts its inflammatory effects as a chemotactic factor by promoting the migration of lymphocytes and neutrophils into inflammatory sites [59,60]. In SARS-CoV-2-infected cells, overproduced PAI-1 binds to TLR4 on macrophages, inducing the secretion of pro-inflammatory cytokines and chemokines [61]. A new vision of PAI-1 that acts as a protective factor during severe sepsis by limiting the inflammatory response has been introduced. Indeed, PAI-1^−/−^ mice showed a reduced level of TNF-α, IL-6, IL-10, and IFN-γ after infection with *B. pseudomallei*, at both the systemic and pulmonary levels [62]. Interestingly, PAI-1^−/−^ mice have an impaired host defense against *Klebsiella*, pneumoccocal pneumonia, or *B. pseudomallei* [62,63,64].

α2-antiplasmin may be considered as a protective mediator by limiting inflammation during severe Gram-negative sepsis, also defined s melioidosis. α2-antiplasmin^−^/^−^ mice showed higher neutrophil numbers, together with the release of pro-inflammatory cytokines [65].

Alpha2-macroglobulin, an inhibitor of plasmin, is able to inhibit several proteases that are released during inflammatory processes. It belongs to the superfamily of acute phase proteins, of which CRP is a well-known example. It seems that alpha2-macroglobulin acts as a pro-resolving mediator in models of systemic inflammation. Interestingly, Dalli et al. have found that the plasma of patients with sepsis contains lipid microparticles enriched with alpha2-macroglobulin produced by activated neutrophils. The administration of alpha2-macroglobulin-enriched-microparticles to mice with sepsis elevated immunoresolvent lipid mediator levels in inflammatory exudates and reduced systemic inflammation [66].

In septic patients, TAFI levels are inversely correlated with inflammation-associated complications [67]. In a septic rat model with *Pseudomonas aeruginosa*, the inhibition of TAFIa attenuated the systemic inflammatory response [68]. TAFI^−^/^−^ mice showed significantly higher concentrations of IL-1beta, TNF-beta, and IL-6, with increased lung infiltration of neutrophils [69]. It is important to underline that TAFI operates on two fronts: on the one hand, it acts as an inhibitor of fibrinolysis by removing C-terminal lysines from fibrin; on the other hand, activated complement factors C3a and C5a can be inactivated by TAFI [70]. In line with these results, Vollrath et al. have recently found decreased TAFI levels and elevated C5a levels in patients with septic complications. The authors concluded that TAFI correlated negatively with anaphilotoxin C5a, suggesting a link between inflammation, complement, and coagulation in sepsis. [71].

### 3.2. Links between Fibrinolytic Mediators and Coagulation Abnormalities in Sepsis

In sepsis, circulating fibrinogen increases as a part of the acute phase response, thus leading to increased total fibrin formation. Thus, plasmin activation initially occurs in response to fibrin formation, as demonstrated by increased PAP complex (a marker of activation of fibrinolysis) and D-dimer (a marker of coagulation with subsequent fibrinolysis) [3]. At a later stage, fibrinolytic activity is inadequate, and there could be many causes. Both excessive consumption of plasminogen and decreased synthesis due to sepsis-associated liver failure could be triggers of fibrinolysis impairment and the hypercoagulable state, but other mechanisms are coming to light. In fact, patients with non-liver sepsis were characterized by a strong hyperfibrinogenemia with a thrombogenic clot structure and a profound resistance to fibrinolysis [72].

The investigation of the effect of cell-free DNA on fibrinolysis in the context of sepsis has provided interesting data. Cell-free DNA can be released through different cellular processes, including apoptosis, necrosis, or via neutrophils, as a component of NETs. Increased levels of cell-free DNA in septic patients downregulated fibrinolysis by inhibiting plasmin-mediated fibrin degradation [73]. It was demonstrated that cell-free DNA incorporated into a clot confers resistance to plasmin-mediated degradation by modifying the structure of fibrin [74].

Another important mechanism underlying the fibrinolytic shutdown and responsible for the impaired dissolution of microthrombi is represented by active elastases in NETs. These active enzymes determine the production of plasminogen fragments with inhibitory activity, which act as competitors of plasminogen binding to fibrin and decrease the plasmin formation [47]. Taken together, these data show that NETs may be responsible for intravascular coagulation by affecting fibrinolytic mediators.

The main inhibitor of fibrinolysis PAI-1 has been extensively investigated as a major player in the hypercoagulable state associated with sepsis. In addition to endothelial cells, platelets are able to release PAI-1 after activation. PAI-1 upregulation leads to coagulopathy characterized by intravascular thrombi in COVID-19. In particular, in SARS-CoV-2-infected cells, a positive feedback loop established between STAT3 and PAI-1 seems to be responsible for thrombosis in COVID-19 [61]. Since plasma levels of PAI-1 significantly correlate with the levels of inflammatory biomarkers, it is conceivable that enhanced inflammatory response may alter the fibrinolytic pathway via serum PAI-1. Interestingly, PAI-1 polymorphism was linked to an increased risk of developing septic shock from meningococcal infection [75].

Sepsis-associated fibrinolytic shutdown could also be mediated by TAFI activity. TAFI is activated by thrombin and thrombin/thrombomodulin complex, but also other proteases, including plasmin and neutrophil elastases [76]. The role of TAFI in the development of coagulopathy abnormalities in sepsis is still unclear. In fact, the studies on the correlation between TAFI levels and sepsis-associated DIC are conflicting. Emonts et al. have found increased levels of TAFI in septic patients, and TAFI activation led to the thrombogenic pathway [77]. Instead, Haiakawa et al. demonstrated that TAFI antigen and its activity were drastically reduced in patients with sepsis-associated DIC. Several mechanisms could be involved in this impairment, such as decreased synthesis and/or increased clearance. However, the most reliable explanation is that excessive thrombin activation leads to the consumption of TAFI [78].

Hence, fibrinolytic shutdown during sepsis can be caused by both PAI-1 and TAFI. Zeerlender et al. speculated that PAI-1 is released early by the endothelium in sepsis, whereas TAFI might contribute to the inhibition of fibrinolysis in later stages of sepsis [79].

The procoagulant state associated with septic shock could be counterbalanced by cell-derived microvesicles released by leukocytes and endothelial cells. These microvesicles exhibit fibrinolytic properties via their plasmin-generation capacity. The binding and subsequent activation of plasminogen by leukocyte-derived microvesicles is mediated by the uPA/uPAR system, whereas endothelial microvesicles utilize tPA [80]. Septic shock patients with high levels of microvesicle-dependent plasmin-generation capacity exhibit a higher survival than patients with a lower level of these microvesicles [81]. Furthermore, microvesicles derived from granulocytes of septic shock patients ensure uPA-dependent plasminogen activation, thus promoting clot lysis [82].

In summary, it is clear that the hypercoagulable state in septic patients is multifactorial and complex. The expression of tissue factor and increased thrombin generation leads to the persistent activation of coagulation, which is potentiated by a state of fibrinolysis shutdown, mediated by a significant increase in PAI-1 and TAFI.

### 3.3. Links between Fibrinolytic Mediators and Endothelial Injury in Sepsis

Sepsis induces a shift in endothelial cell profile toward the anti-fibrinolytic phenotype. In particular, septic endothelial cells are characterized by the over-expression of *SERPINE-1*, the gene encoding PAI-1, and plasma levels of PAI-1 correlate with disease severity [83]. An in vitro study has shown that IL-6 trans-signaling is critical for the production of PAI-1 in human umbilical vein endothelial cells (HUVECs). PAI-1 release is blocked by treatment with tocilizumab, an IL-6 receptor antagonist [84]. Elevated PAI-1 levels, mainly released from endothelium, inhibit both uPA and tPA, thus impairing fibrinolysis. The mechanisms underlying this pathogenetic process are partially known. PAI-1 is able to bind to uPA/uPAR complex, and the complex including uPA-uPAR-PAI-1 undergoes rapid internalization and degradation [85]. Furthermore, uPA-uPAR-PAI-1 complexes are shed by endothelial cells in a soluble form or in the form of microparticles. Notably, PAI-1 induces the procoagulant activity of endothelial microparticles through thrombin formation that in turn stimulates PAI-1 synthesis, thus generating an auto-amplification loop [86].

A large body of evidence has shown that upon injury, PAI-1 increase is associated with a marked release of tPA from the endothelium into circulation, which, however, is not enough to avoid a fibrinolytic shutdown in endotoxemia or sepsis [87].

Interestingly, the lung endothelial expression of uPA and tPA is differentially regulated; at baseline, endothelial cells from the pulmonary artery express tPA, whereas endothelial cells from the microvascular pulmonary circulation over-express uPA. In response to inflammatory stimuli, uPA expression increases, tPA is downregulated, and PAI-1 is upregulated, in all endothelial cell types [88]. This response suggests that the fibrinolytic activity is mainly regulated by tPA, whereas uPA is predominantly involved in other cell activities independent from fibrinolysis.

Recently, it has been demonstrated that uPA exerts a therapeutic effect against sepsis by enhancing cell viability and reducing apoptosis in septic HUVECs. Notably, uPA seems not to be involved in the regulation of inflammatory cytokine expression in septic endothelial cells [89].

The involvement of the uPAR system in sepsis-associated vascular dysfunction could be mediated by uPAR/FPRs crosstalk on the endothelial cell surface. In fact, a proteolytic cleavage of uPAR operated by different proteases, including uPA itself, and/or conformational change by binding to uPA determines the exposure of sequence SRSRY (aa 88–92) at its N-terminus. This sequence is responsible for uPAR interaction with FPRs and is involved in the pathogenesis of fibrotic diseases, including systemic sclerosis [39,41]. Starting from the evidence that endothelial FPRs are key contributors to impaired barrier function in SIRS and sepsis patients [32], it is conceivable that uPAR may be responsible for enhanced FPR activation.

Taken together, this evidence indicates that alterations in PAS occur as part of the physiological response to infection; however, as sepsis develops, these modifications contribute to microthrombus formation and poor survival.

Figure 1 illustrates the succession of pathological alterations in fibrinolysis during sepsis described above.

## 4. Marker of Fibrinolytic Activity as a Useful Approach in the Routine Diagnosis of Sepsis

The COVID-19 pandemic has highlighted the importance of having biomarkers available for the early diagnosis, management, and prognosis of sepsis, as well as for the recognition of complications. Several biomarkers have been evaluated over the last few decades in the context of the diagnosis, prognosis, and assessment of therapy in sepsis, but guidance on how to optimize their use in clinical practice is necessary.

The clinical value of fibrinolysis in sepsis is now known; therefore, laboratory tests able to detect the components of PAS could be a useful approach to the management of patients with sepsis.

In this section, the main PAS-associated biomarkers with clinical significance are described in order to evaluate their application in the clinical practice.

### 4.1. PAI-1

Fibrinolysis shutdown and multiple microthrombi occurring in sepsis impair microcirculation, thus leading to organ failure. The PAI-1 level reflects coagulo-fibrinolytic abnormalities and, thus, may be a useful marker to assess the severity of sepsis or prevent sepsis-induced organ failure.

Plasma levels of PAI-1 increase before fibrin formation, and for this reason, it is considered a sensitive marker that detects the phase of the coagulopathy before the DIC condition. In vivo studies have demonstrated that PAI-1 levels increase 2 h after the challenge of both baboons with *Escherichia Coli* and healthy humans with endotoxin, thus confirming the early detection of this marker [90,91].

The plasma concentration of PAI-1 is low (20 ng/mL), and the active form is stabilized by binding to vitronectin. There is a large variation in antigen levels in normal subjects (1–40 ng/mL), and it can vary in different disease states and conditions such as obesity. However, several clinical studies have demonstrated the diagnostic validity of PAI-1 in sepsis. Hoshino et al. showed that PAI-1 is the most predictive marker of 28-day mortality among sepsis biomarkers, and the cutoff value, determined by ROC curve analysis, is 83 ng/mL. Moreover, PAI-1 is superior to the SOFA score for predicting mortality [92]. Tipoe et al. have conducted a systematic review and meta-analysis of the predictive value of PAI-1 in sepsis, thereby confirming the relationships between PAI-1 plasma levels and disease severity or mortality in sepsis [93]. During the COVID-19 pandemic, much attention has been paid to the measurement of PAI-1 plasma levels as a predictive marker of prothrombotic events. Zuo et al. have found that elevated PAI-1 levels were associated with a worse respiratory condition [94]. Since the induction of PAI-1 occurs during the moderate stage before COVID-19 patients require intensive care, the evaluation of PAI-1 would help in designing better thromboprophylaxis to limit the disease severity [95].

The DNA genotyping of PAI-1 could in the future represent a valid alternative to the measurement of PAI-1 plasma levels. Despite contrasting data in the literature, Lorente et al. have found that the PAI-1 4G/4G genotype is associated with high plasma levels of PAI-1 and mortality in patients with sepsis [96]. A meta-analysis suggests that the PAI-1 4G/5G polymorphisms may be risk factors for the development of pediatric sepsis [97]. Finally, the determination of PAI-1 polymorphisms could help in the selection of patients with a higher mortality risk.

### 4.2. α2-Antiplasmin

Native α2-antiplasmin circulates at a concentration of approximately 70 mg/mL and is primarily synthesized and secreted by the liver but also by the kidney and the brain. Upon its synthesis, α2-antiplasmin is enzymatically modified in the circulation at both the N- and the C-terminal, which affects its fibrin-crosslinking and plasmin-binding abilities, respectively [98]. The mean value of α2-antiplasmin in the plasma of healthy subjects is 0.8–1.2 IU/mL [99].

α2-antiplasmin represents the primary and fast inhibitor of plasmin, forming a 1:1 stable complex with plasmin (PAP complex). The presence of PAP complex in the blood indicates plasminogen activation [100].

To date, there are still conflicting data on the serum concentration of α2-antiplasmin in patients with sepsis. Recently, it has been found that α2-antiplasmin activity is notably higher in patients with severe COVID-19 than in patients with sepsis, suggesting a difference between the endotheliopathy occurring in COVID-19 and sepsis [101]. More interestingly, plasma samples from long COVID-19 contain large anomalous deposits that are resistant to fibrinolysis. Proteomic analysis has shown that α2-antiplasmin is trapped in these deposits [102]. Moreover, patients with severe Gram-negative sepsis showed elevated α2-antiplasmin plasma levels [65].

The data on circulating levels of PAP complex are truly contradictory. However, several studies have demonstrated elevated circulating PAP complex in patients with sepsis [103]. High levels of PAP complex were associated with increased mortality risk [104,105]. Instead, Semeraro et al. have found that low PAP complex combined with thrombocytopenia was associated with high mortality [106].

Notably, the balance of coagulation and fibrinolysis can be measured using the TAT/PAP ratio. The elevated TAT/PAP ratio from the imbalance between coagulation and fibrinolysis led to the onset of organ dysfunction [107].

Compared to hemostatic parameters, including fibrin degradation products (FDP), D-dimer, and PAI-1, PAP complex analysis has been known to be a more useful parameter in the evaluation of DIC and sepsis severity [108].

Finally, no standardization of methods for detecting α2-antiplasmin and result interpretation has been found. Since the activity of α2-antiplasmin is modified by N- and C-terminal proteolytic cleavage, the development of tests capable of detecting these modifications could be a valuable aid to better understand sepsis-associated fibrinolytic shutdown and improve the panel of biomarkers to predict, identify, or provide new approaches to treat sepsis.

### 4.3. Plasminogen Activators: tPA and uPA

tPA could be considered a marker of endothelial activation. Mavrommatis et al. have found elevated tPA levels in patients with both sepsis and septic shock, with a simultaneous increase in PAI-1. These data suggest that the excess of tPA is inactivated by increased PAI-1 [103]. In the Protocolized Care in Early Septic Shock (ProCESS) trial, increased circulating tPA at admission was associated with increased mortality [109]. Elevated levels of tPA levels were also reported in COVID-19 patients, with hyperfibrinolysis associated with higher tPA levels and a poor outcome [94]. Some authors have found reduced tPA expression and/or release or no association with mortality. tPA lysis time may be utilized to analyze the mechanisms of fibrinolysis impairment and guide corrective therapies in the critically ill, as suggested by Coupland et al. [110]. Finally, it should be understood whether there are any differences between the concentration of tPA and its activity, as the ability to activate fibrinolysis is influenced by high concentrations of PAI-1.

The role of uPA as a laboratory parameter in the clinical management of sepsis remains to be determined. More recently, it has been shown that the plasma levels of uPA were significantly increased in sepsis patients compared with SIRS patients, and uPA exhibits significant diagnostic value in patients with sepsis [111]. In another study that enrolled 81 patients with septic shock due to pneumonia, plasma uPA was associated with organ dysfunction and mortality [112].

The presence of a higher level of uPA in plasma samples from patients with sepsis could be considered a protective factor against sepsis, as uPA is reported to be capable of regulating inflammation and immune responses and inducing plasminogen activation. Circulating uPA can be inactivated by suPAR by acting as a uPA scavenger, as demonstrated by Wilhem et al. in human cancer cells [113].

Although the validity of uPA as a sepsis marker remains to be clarified, its role in predicting mortality and organ dysfunction looks promising.

### 4.4. suPAR

suPAR has great potential to improve the diagnosis and treatment of sepsis. During an inflammatory response, uPAR expressed on the membrane of immune and endothelial cells is shed and circulates in plasma as a soluble form, abbreviated as suPAR. Beyond plasma, suPAR is detectable in other different body fluids, including serum, urine, sputum, pleural effusion, peritoneal fluid, and cerebrospinal fluid. Most of the data available in the literature concern the suPAR level in the serum.

The clinical significance of suPAR in sepsis has been shown in several studies. A recent systematic review and meta-analysis has compared the clinical guiding value of suPAR and procalcitonin, which is the most used marker in sepsis. Therefore, suPAR exhibits higher specificity and is able to differentiate sespis from SIRS. Patients with urosepsis and endotoxemia had elevated levels of suPAR in both plasma and urine [114]. suPAR is also considered as a marker of case fatality in patients with bacteremia [115].

A position paper by the Hellenic Sepsis Study Group has suggested that suPAR levels > 6 ng/mL are an alarming sign of risk for unfavorable outcomes in the Emergency Department [116]. In particular, suPAR cutoffs of below 4, between 4 and 6, and above 6 ng/mL can identify acute medical patients who have a low, medium, or high risk of mortality [117].

The COVID-19 pandemic has underlined the importance of introducing suPAR into clinical practice. Napolitano et al. have proposed suPAR as a serum biomarker of clinical severity and outcome in hospitalized patients with COVID-19 [42]. The SAVE-MORE study (suPAR-guided Anakinra treatment for Validation of the risk and Early Management Of seveRE respiratory failure by COVID-19) was a clinical trial that relies on the early identification of patients at risk for unfavorable outcomes using suPAR and the provision of targeted treatment with anakinra. The results from this study showed that an early start to treatment with anakinra guided by suPAR levels in hospitalized patients reduced the risk of worse clinical outcomes [117].

Since serum suPAR is associated with sepsis severity and 28-day mortality, a recent work showed that adding suPAR to qSOFA increased the ROC curve area, suggesting that this might be a useful tool in sepsis mortality prediction models [118].

Although access to suPAR testing is limited, overall, the data demonstrate that suPAR early predicts the outcome of patients with sepsis, and it may be used for monitoring the response to treatment.

### 4.5. Summary of PAS-Associated Biomarkers in Sepsis: Considerations and Assumptions

Although we still lack diagnostic assays to accurately assess the fibrinolytic changes and to follow them over time, opinions on the diagnostic and prognostic validity of circulating PAS-associated biomarkers are almost all in agreement. In Table 1, we report the main circulating fibrinolytic markers and schematize their potential as prognostic markers for sepsis-associated organ failure and mortality. The studies from which data are extrapolated included severe sepsis patients with blood samples collected at enrolment in the study when they are admitted to the Intensive Care Unit. Studies aimed at examining markers along the course of the disease are very few. Normal range and cutoff value are not available for all markers listed in Table 1.

PAI-1 has consistently been linked to organ dysfunction in sepsis patients. The 4G/4G genotype of PAI-1 polymorphism is associated with higher serum PAI-1 concentrations and mortality in severe sepsis. There are specific commercialized kits to detect the different molecular forms of PAI-1 in blood, including active PAI-1 (complexed with vitronectin), latent PAI-1, and the form complexed with tPA and uPA. We assume that the analysis of active PAI-1 could be more suitable for detecting fibrinolytic activity in sepsis. A2-antiplasmin has been analyzed both as an unbound form and in the complex with plasmin (PAP complex) using ELISA methods. It seems that α2-antiplasmin is protective during Gram-negative sepsis, but further studies are needed.

In contrast to PAI-1, less is known about the markers associated with the activation of PAS during sepsis. However, the clinical significance of measuring plasminogen activators, tPA and uPA, has been introduced by recent studies regarding the fibrinolytic system. Sensitive ELISAs are available for the measurements of tPA and uPA, but alternative assays are also used in the literature, including tPA lysis time and uPA activity assays. Elevated levels of plasminogen activators has been associated with worse outcome in sepsis patients, but there are conflicting data on the clinical significance of uPA. In fact, it would seem that uPA is a protective factor rather than a marker of damage.

Finally, the most interesting marker as a prognostic factor in sepsis is suPAR. Detectable with the ELISA method, suPAR achieves better performance in clinical practice both in predicting mortality and as a guide to therapy.

## 5. Discussion

The three typical hallmarks of sepsis are inflammation, hypercoagulable state, and endothelial dysfunction. In this review, we have illustrated the pathogenetic contributions from activators, inhibitors, and protein components of PAS in the three sepsis hallmarks.

Upon endothelial damage, elevated levels of PAI-1 are released from endothelial cells and activated platelets [119]. Beyond its ability to inhibit plasminogen activation, PAI-1 is able to activate TLRs expressed on the monocyte surface, thus contributing to cytokine release and acting as a pro-inflammatory mediator. Furthermore, PAI-1 acts as a chemotactic factor promoting the migration of lymphocytes and neutrophils into the inflammatory site.

Endothelial cells and platelets can release uPA, which catalyzes the conversion of plasminogen into plasmin. On the other hand, the effect of uPA is independent of plasmin generation, as its interaction with uPAR is linked to inflammation and innate immune activation. uPAR, expressed on neutrophils, monocytes, macrophages, and endothelial cells, contributes to cytokine storm and promotes C5a signaling, thus inducing neutrophil activation. Furthermore, uPAR interacts with chemotactic receptors of the FPR family that are implicated in ROS generation, degranulation, extravasation, and the migration of neutrophils.

A number of proteases including chemotrypsin, trypsin, and uPA itself, can cleave uPAR, thus exposing a chemotactic epitope able to activate chemotaxis receptors of the FPR family on inflammatory cells. Moreover, various types of phospholipases can mediate uPAR release from the cell surface with the release of soluble forms of the receptor, known as suPAR. Serum suPAR is dramatically elevated during sepsis and represents a potential biomarker for predicting its severity better than many other biomarkers, such as CRP and PCT. Our study and other studies indicate that serum suPAR levels correlate with different disease severity in COVID-19 patients to support the clinical value of this biomarker as a prognostic index.

tPA is constitutively released from endothelial cells, but a second pool of tPA is contained within storage vehicles of endothelial cells and released in response to various stimuli. Microvesicles that utilize tPA and/or uPA can counterbalance the procoagulant state associated with septic shock and sepsis.

Neutrophils play a crucial role in the pathogenesis of sepsis; in particular, they exert pro-thrombogenic effects through PAS inhibition performed by NETs. NET-derived elastases produce plasminogen fragments that act as competitors of plasminogen binding to fibrin, thus inhibiting plasmin generation. Cell-free DNA derived from NETs is able to modify the fibrin structure, conferring resistance to plasmin activity.

We can hypothesize that in the first phase of sepsis, plasmin activation is predominant due to the increased fibrin formation. In this stage, PAS can induce pathological alterations through its pro-inflammatory effects. In the later stages, plasminogen consumption combined with liver dysfunction leads to low circulating plasminogen levels, which cause fibrinolysis impairment, also defined as fibrinolysis shutdown.

Marked activation of coagulation is a common pathogenetic factor among DIC types. Unlike DIC occurring in cancer or in bleeding disorders, sepsis shows suppressed fibrinolytic-type DIC. For these reasons, the evaluation of circulating PAI-1 should be conducted in clinical practice in order to quantify the impairment of the fibrinolytic pathway.

The present review also provides an overview of potential laboratory markers of altered PAS in sepsis and discusses future perspectives for the diagnosis and prognosis of this complex disease condition. To summarize, PAI-1 has been linked to DIC and increased mortality in patients with sepsis. Although the role of plasminogen activators in sepsis remains to be clarified, suPAR may be considered as a prognostic and/or predictive biomarker that helps guide the therapies, as already established during the COVID-19 pandemic.

## 6. Conclusions

Sepsis is a complex and intricate pathological condition that cannot be simply described by the dysregulation of a single hemostatic pathway or by a single biological parameter. A wide range of metabolites associated with inflammatory cascades and cellular components are being investigated, but no marker has been validated. Moreover, the use of approved scoring systems to help with decision making for sepsis patients is complicated. However, the implementation of PAS-associated markers in routine laboratory work, in addition to the current clinical predictive models, could considerably enhance the sensitivity of the detection of critical illness. Unlike common diagnostic coagulation tests, methods for detecting PAS-associated mediators are poorly standardized. The timing of blood sampling is important, as many of PAS markers show changes during sepsis.

Despite the difficulties of methods for studying fibrinolysis, research on the role of PAS in sepsis may lead to better diagnostic techniques and therapeutics.

## Figures and Tables

**Figure 1 ijms-24-12376-f001:**
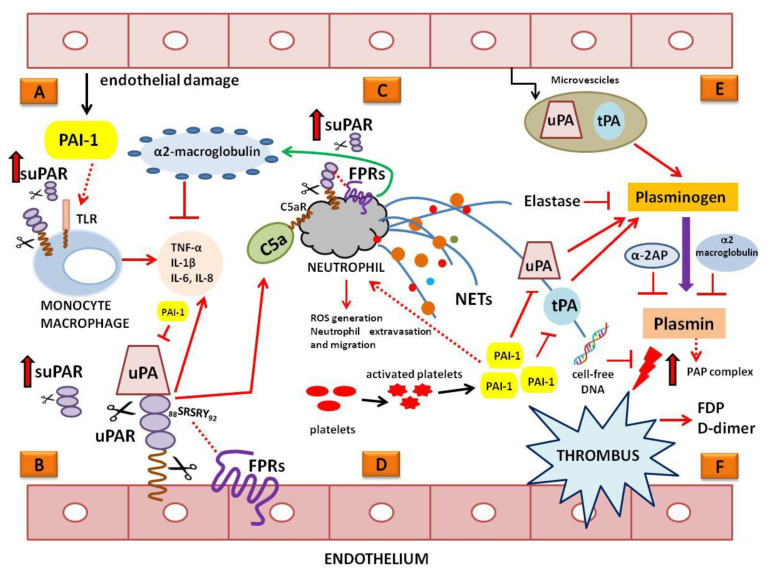
Role of the plasminogen activation system (PAS) in the pathogenesis of sepsis. (**A**) Beyond its well-established role in the inhibition of plasminogen activation, PAI-1 binds to TLRs on macrophages, inducing the release of pro-inflammatory cytokines, including TNF-α, IL-6, IL-10, and IFN-γ. (**B**) Together with tPA, uPA is a main activator of plasminogen. Besides its involvement in plasminogen activation, uPA induces intracellular signal transduction through interaction with its specific receptor uPAR. uPAR, expressed on endothelial and immune cells, is linked to inflammation and innate immune activation. It acts as a part of PRR signaling to PAMPs and DAMPs by reinforcing TLR signaling. uPAR can be shed from the cell surface (suPAR) and is detectable in multiple body fluids. Upon exposure to chemotactic sequence (_88_SRSRY_92_), both cell-surface-associated and soluble uPAR forms can interact with N-formyl peptide receptors (FPRs), thus amplifying inflammation, the generation of reactive oxygen species (ROSs), neutrophil extravasation, and migration. uPAR is also implicated in the activation of C5a, which elicits a severe inflammatory response. (**C**) The activation of neutrophils causes the release of neutrophil extracellular traps (NETs). NETs contribute to sepsis-associated inflammation and thrombosis; DNA-bound elastase is able to degrade plasminogen, thus contributing to its consumption. Cell-free DNA blocks plasmin-mediated fibrin degradation. The plasma of patients with sepsis contains lipid-microparticle-enriched α2-macroglobulin produced by activated neutrophils. (**D**) Activated platelets contribute to the release of high amounts of PAI-1, which promotes neutrophil migration and contributes to the fibrinolytic shutdown. (**E**) The procoagulant state could be reversed by microvesicles with fibrinolytic property released by leukocytes and endothelial cells. These microvesicles probably contains plasminogen activators and are able to generate plasmin. (**F**) Plasmin is negatively regulated by α2-antiplasmin (α2-AP) and α2-macroglobulin. Plasmin/α2-AP (PAP) complex is considered as a marker of fibrinolytic activation in response to fibrin formation. Fibrin-degradation products (FDP) and D-dimer indicate the amount of breakdown of clots. ⊥ indicate inhibitory signals; the red dashed line indicates the specific interaction between FPRs and uPAR; both dashed and continuous red arrows indicate the contacts between the molecular and cellular pathways; the red arrows with black borders indicate the increase of suPAR; the green arrow indicate the ability of neutrophils to release lipid-microparticle-enriched α2-macroglobulin; lightning symbol indicate the ability of plasmin to dissolve intravascular thrombi into soluble parts.

**Table 1 ijms-24-12376-t001:** PAS-associated biomarker in sepsis.

Marker	Laboratory Methods	Normal Range	Cutoff Value	Timing	Clinical Significance
PAI-1	ELISA for different molecular forms in blood *	1–40 ng/mL	83 ng/mL	Early detection Decline in survivors	DIC: ↑ Organ failure: ↑ Severe COVID-19: ↑ Mortality: ↑
PAI-1 genotype	Allele-specific PCR	-	-	-	Mortality: 4G/4G polymorphism Pediatric sepsis: 4G/5G polymorphism
α2-AP	ELISA	0.8–1.2 IU/mL	-	Persistence of Long COVID-10	Severe COVID-19: ↑ Protective during Gram-Negative sepsis
PAP complex	ELISA	-	-	-	DIC: ↑ Organ failure: ↑ Mortality: ↑
tPA	ELISA Lysis time test	1–5 ng/mL	-	Decline in survivors	Mortality: ↑ Severe COVID-19: ↑
uPA	uPA activity assay ELISA	-	-	-	Organ failure: ↑ Mortality: ↑ Protective factor?
suPAR	ELISA	2–3 ng/mL	4 ng/mL: low risk 4–6 ng/mL: medium risk 6 ng/mL: high risk	Early identification of patients with adverse outcome	Organ failure: ↑ Mortality: ↑ Targeted treatment

Symbol: ↑ indicate increased levels of specific marker associated with disease severity. Abbreviations: PAI-1, plasminogen activator inhibitor type 1; DIC, disseminated intravascular coagulation; α2-AP,α2 antiplasmin; PAP, plasmin- α2 antiplasmin; tPA, tissue-type Plasminogen Activator; uPA, urokinase Plasminogen Activator; suPAR, soluble urokinase plasminogen activator receptor. ***** The molecular forms of PAI-1 in blood are active PAI-1 (complexed with vitronectin), latent PAI-1, and the form complexed with tPA and uPA.

## Data Availability

Data is contained within the article.

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
