# Peer review of "Plasminogen System in the Pathophysiology of Sepsis: Upcoming Biomarkers"

_ijms, 2023, doi:10.3390/ijms241512376_

Round 1

Reviewer 1 Report

The paper addresses the main question they posed, but no hypothesis
formulation is necessary for this type of research publication. The topic is highly interesting for sepsis researchers working with translational questions in the field, but not original. It adds a fine narrative review of the existing literature evidence. The paper is well written. The text is clear and easy to read, but an addition of a Table with key messages/main conclusions should be considered. The manuscript involves a hot topic of translational sepsis research in a well-written manner. Minor remarks:

1. "Sepsis" does not need to capitalized in every sentence.

2. Please add a Table for concluding the main messages of the manuscript.

Reviewer 2 Report

The review aims to summarize the current knowledge on the role of the plasminogen activation system (PAS) in the pathology of sepsis. Its clinical utility for diagnosis and prognosis of patients with sepsis. The review is clearly presented, compelling, and provides useful information. Both inflammatory/infectious diseases and coagulopathy fields of research are to benefit from this work.

One suggestion to further enhance the work would be to illustrate the ideal timing of the different assays (i.e., for the PAS biomarker discussed in the review). This could be an illustration depicting time after a ‘challenge’, the biomarker to test, normal range, and cut-off.
